# The Neutron Absorption Capacity of a Composite Material Based on Ultrahigh Molecular Weight Polyethylene Under Reactor Radiation Conditions

**DOI:** 10.3390/polym16233425

**Published:** 2024-12-05

**Authors:** Mazhyn Skakov, Baurzhan Tuyakbayev, Yernat Kozhakhmetov, Yerzhan Sapatayev

**Affiliations:** 1NJSC S., Amanzholov East Kazakhstan University, Ust-Kamenogorsk 070000, Kazakhstan; skakovnnc@gmail.com; 2NJSC D., Serikbayev East Kazakhstan Technical University, Ust-Kamenogorsk 070000, Kazakhstan; ykozhakhmetov@ektu.kz; 3Institute of Atomic Energy NNC RK, Kurchatov 140000, Kazakhstan; sapatayev@nnc.kz

**Keywords:** radiation-protective material, ultra-high molecular weight polyethylene, reactor irradiation, composite material

## Abstract

This work presents the results of a study on the influence of fillers on the neutron absorption capacity of materials made from ultra-high molecular weight polyethylene (UHMWPE). Composite materials based on UHMWPE were obtained using gas-flame technology with the addition of powdered UHMWPE fillers (H_3_BO_3_, WC, and PbO). A radiation cassette has been developed and constructed for conducting studies on the neutron absorption capacity of the material, allowing for the placement of a sample with activation indicators. Samples of UHMWPE with fillers were irradiated at different doses on the unique research reactor IVG-1M, located at the National Nuclear Center of the Republic of Kazakhstan in the city of Kurchatov. The reaction rate of ^63^Cu (n, g), ^64^Cu and ^58^Ni (n, p)^58^Co on activation indicators and neutron flux density at the sample location were determined. Neutron-physical and thermal-physical calculations were performed in order to determine their characteristics. The structure and phase state of UHMWPE with fillers were studied before and after irradiation.

## 1. Introduction

The usage of polymeric materials and composites for protection against various types of radiation has become quite widespread [1]. The most effective means of protection against fast neutrons is to slow them down, since the lower the energy of the neutrons, the higher their permissible threshold [2]. Many researchers have developed polymer composites specifically for their radiation-protective properties. The reaction of radiation capture of neutrons by the isotope boron 10B is often used to absorb slow neutrons. The content of the isotope boron 10B in the natural mixture of isotopes is 19.8%. The introduction of boron into the polymer is carried out in the form of various boron-containing compounds. Shielding from gamma rays formed as a result of neutron capture with a significant weakening of neutron fluxes becomes more important [3].

Today, the industry produces commercially successful slabs and blocks of borated polyethylene designed to slow down and absorb neutrons [4]. The slabs and blocks are used in structures operating at temperatures up to 130 °C and relative air humidity up to 100%, as well as in reactor irradiation conditions. It is permissible to use the slabs in water, including sea water, at temperatures from −2 °C to +35 °C. Borated slabs are made from a mixture of high-pressure polyethylene with amorphous boron. Depending on the percentage of boron content, slabs of various grades are produced [5]. The combination of a hydrogen-containing polymer and boron carbide in one material allows for effective radiation protection not only from thermal neutrons but also from fast neutrons, since fast neutrons, as a result of elastic scattering on the hydrogen of polyethylene, release energy to thermal values and are absorbed by boron [6,7].

Modern nuclear energy and radiation protection technologies place high demands on the materials used in reactor systems. One of the key characteristics of such materials is their ability to absorb neutrons, which is critical to ensuring the safety and efficiency of nuclear reactors [8]. In this regard, composite materials based on ultra-high molecular weight polyethylene (UHMWPE) are of particular interest due to their unique properties, such as high strength, heat resistance, and excellent radiation characteristics.

Ultra-high molecular weight polyethylene, due to its molecular structure and high mechanical properties, has long been used in various fields, including medicine, aerospace, and, of course, nuclear energy. In combination with various fillers, such as boric acid (H_3_BO_3_), tungsten carbide (WC), and lead oxide (PbO), it demonstrates increased efficiency in neutron absorption, which is especially important for neutron flux control and radiation protection in nuclear power engineering. Boric acid is known for its ability to absorb thermal neutrons; tungsten carbide is effective in absorbing fast neutrons; and lead oxide can play a role in gamma radiation protection. Thus, the combination of these fillers with the UHMWPE matrix can lead to the creation of materials with improved neutron absorption properties.

In connection with the above, the purpose of this work is to determine the ability of a composite material obtained by the gas flame method, based on UHMWPE with fillers, to absorb neutron fluxes of the reactor radiation spectrum.

However, the effect of reactor irradiation on the neutron absorption properties of UHMWPE composite materials requires detailed study. Reactor radiation can lead to various changes in the structure and composition of the material, which in turn can affect its absorption properties and durability. An important aspect is also the assessment of the stability of the material under long-term exposure to neutron flux and its ability to retain its original properties under real operational loads.

This study is aimed at assessing the effect of reactor irradiation on the neutron absorption characteristics of a composite material based on UHMWPE with H_3_BO_3_, WC, and PbO fillers. The work involves complex experiments and analyses, including full-scale neutron experiments at the unique IVG.1M research reactor of the NNC RK. The work involves an analysis of changes occurring in the material under the influence of neutron radiation and an assessment of its properties before and after irradiation. The results obtained in this work are critical for optimizing the composition of composite materials and increasing their efficiency under real operating conditions.

Studying the effects of reactor irradiation on such composite materials not only contributes to the development of more effective radiation shielding materials but also helps to ensure safer and more efficient work of personnel involved in nuclear power, which is an important step for further development.

## 2. Materials and Methods

The material chosen for the study was a composite material based on ultra-high molecular weight polyethylene with fillers (H_3_BO_3_, WC, PbO), produced by Nantong Yangba Polyethylene Co., Ltd., Nantong, China. UHMWPE is a white powder with an average particle size of 150 μm, Tm = 135 °C, density of 930 kg/m^3^, molecular weight of 2·10^6^ mol^−1^, bulk density >0.4 g/cm^3^. The fillers of the composite material were H_3_BO_3_ powders with a molar mass of 61.83 g/mol in accordance with [9], WC with a bulk density of 15.77 g/cm^3^ in accordance with [10] with an average particle size of 60 μm, and PbO with a molar mass of 239.1988 g/mol and a density of 9.38 g/cm^3^.

The content of fillers and base in percentage by weight is indicated in Table 1.

Composite materials based on UHMWPE were obtained using the flame spraying technology previously developed by the authors [11]. The process of synthesis of polymer composites included the following stages: preliminary drying of filler powders at a temperature of 50–100 °C, after which all components were thoroughly mixed in a roller mill. A mixture of ultra-high molecular weight polyethylene (50 wt.%), tungsten carbide (20 wt.%), boric acid (20 wt.%), and lead oxide (10 wt.%) was used as a filler. Before the spraying process, the substrate was preheated to a temperature of 95–100 °C, which ensured optimal conditions for applying the coating with flame spraying.

After completion of the experimental work, three cylindrical samples with a diameter of 20 mm and a height of 25 mm, weighing 10 g each, were made.

Before irradiation, holes with a diameter of 1.5 mm were drilled in the center of each sample to install activation indicators (AI) (Figure 1b) [12]. Nickel and copper wires, designed to determine the degree of neutron flux attenuation, were used as AI. The diameter of the wire AI was 1 mm, and the length was 20 mm.

In addition, the AIs were installed on the side surface of the UHMWPE samples, and copper AIs were installed on the outer surface and along the axis of the extension of the physical experimental channel (PEC).

The irradiated UHMWPE samples were placed in the pockets of a cartridge belt (Figure 1a), installed on the outer surface of the physical experimental channel (PEC). Each pocket contained three samples, into which copper activation indicators (AI1) were installed. In addition, copper AI (AI2) was also installed on the outer surface of the samples, as shown in Figure 1b. The composite material samples were subjected to reactor irradiation in the IVG1M reactor. The irradiation process was carried out at a temperature not exceeding 40 °C. The cartridge belt with the studied samples (Figure 2a) was placed in the IVG1M reactor (Figure 2b), providing the necessary experimental conditions.

When the reactor was operating at nominal power, the samples were exposed to the neutron flux of the reactor spectrum, as well as to “hard” gamma quanta. The IVG.1M nuclear reactor is a research water-cooled, heterogeneous thermal neutron reactor with a beryllium reflector [13]. The main technical characteristics of the IVG.1M research reactor are presented in Table 2.

After irradiation at the reactor, the samples were transferred to a special room in the reactor building to measure the activity of the radioactive isotopes and the activity of the samples themselves in accordance with the well-known activation analysis technique [13].

The structure of irradiated and non-irradiated samples of the composite material was studied using an Xpert PRO X-ray diffractometer, manufactured by PANalytical, Almelo, Netherlands. The exposure time (time per step) during shooting was 61.2 s, the scanning step size for diffraction patterns was 0.026°2θ, and the studied angular range was 5–20°2θ. The operating mode of the PIXcel1D detector is a scanning line detector. Radiation: Cu Kα; voltage and current: 45 kV, 40 mA. A fixed divergence slit with an angular divergence of 1°, an antiscattering slit of 2°, and an incident beam mask marked 10, providing an incident beam width of 9.9 mm, were used. The phase composition and degree of crystallinity were determined in accordance with the recommendations [14,15].

The fractography of the structure and elemental analysis of the irradiated and non-irradiated samples were studied on a Tescan Vega 3 electron microscope using a secondary electron detector. The samples were sputtered with carbon to reduce the effect of charge accumulation on the surfaces of the fragments using a JEOL JEC-560 automatic sputter coater, manufactured by JEOL Ltd., Tokyo, Japan. 

## 3. Results

### 3.1. Neutron and Thermophysical Calculations

In order to justify the safety of reactor experiments and select the optimal irradiation mode for samples, neutron-physical and thermal-physical calculations were performed for a cartridge belt with samples placed above the experimental channel of the IVG.1M reactor.

The purpose of neutron-physical calculations was to determine the energy release in the samples during their irradiation, which allows us to estimate the intensity of neutron impact and potential changes in the material. Thermophysical calculations were aimed at determining the temperature field in the package with samples, which is critical for maintaining safe temperature conditions during irradiation and preventing undesirable thermal effects.

Neutron-physical calculations were carried out using the MCNP5 software [16], which is a universal program for solving problems of radiation transfer in arbitrary three-dimensional geometry with ENDF/B-5,6 constant libraries.

The temperature field of the package with samples was determined using the ANSYS Fluent Release 2021 R2 software package [17].

The temperature distribution of the calculation model was determined by the energy release in the samples and convective heat exchange with the environment.

The condition of natural convection was set on the outer side of the calculation model, while the convection coefficient was taken to be 5 W/(m^2^ × °C), the ambient water temperature was 40 °C.

The properties of the model materials were taken from reference literature and are specified as a functional dependence on temperature [18], the properties of the samples were determined by the mass fractions of the composition (Table 1). Since the calculation had involved low energy values, a double-precision solver was used to obtain a more accurate result. The software package performed the calculation using an iterative method, the energy convergence criterion for the obtained solution was set at 10^−8^ W. The calculated temperature field of the model is shown in Figure 3a, the calculated field of the cartridge belt cross-section at the level of the mid-height of the samples is shown in Figure 3b.

### 3.2. Reactor Irradiation

To ensure the complex effect of reactor irradiation on the structure and properties of UHMWPE with fillers, 4 starts were carried out.

At the first start, one power level of 3 MW and a duration of 0.5 h were implemented, the diagram of which is shown in Figure 4, with an integral irradiation dose of 7.2 × 10^15^ n/cm^2^.

At the second start, three reactor power levels were implemented (P1 = 1 MW (71 min), P2 = 3 MW (71 min), P3 = 6 MW (71 min)) with an integral power of 11.83 MW × h, the diagram of which is shown in Figure 5, with an integral irradiation dose of 5.1 × 10^16^ n/cm^2^.

During the third launch, two reactor power levels were implemented (P1 = 1 MW (71 min), P2 = 3 MW (71 min)) with an integral power of 4.73 MW × h, the diagram of which is shown in Figure 6, with an integral radiation dose of 3.4 × 10^16^ n/cm^2^.

During the fourth launch, four reactor power levels were implemented (P1 = 1 MW (41 min), P2 = 3 MW (41 min), P3 = 6 MW (71 min), P4 = 10 MW (31 min)) with an integral power of 15 MW × h, the diagram of which is shown in Figure 7, with an integral radiation dose of 4.4 × 10^16^ n/cm^2^.

Thus, the studied samples were exposed to the neutron study of the reactor spectrum with an integral dose of irradiation: for all 4 irradiation modes. The total dose of irradiation received by the samples after each realized start-up was: after the first start-up 7.2 × 10^15^ n/cm^2^, after the second start-up 5.8 × 10^16^ n/cm^2^, after the third start-up 9.3 × 10^16^ n/cm^2^, after the fourth start-up 1.4 × 10^17^ n/cm^2^.

### 3.3. Results of Activation Analysis and Neutron Absorption Coefficients of Composite Material Based on UHMWPE

After irradiation at the reactor, the samples were transferred to a special room in the reactor building to measure the activity of the irradiated samples and the samples themselves. The gamma spectra of the irradiated samples were measured after they were kept for more than 22 h. The activity of the activation products ^64^Cu, ^58^Co, formed in the reactions ^64^Cu (n,γ) ^64^Cu, was calculated at the time of completion of the start-up using Equation (1) [19,20]:(1)$$A_{t}=\frac{kS}{t_{l}\varepsilon \eta e^{-\lambda t_{e}}}$$
where,

S—the area of the peak of total absorption with energy E = 511, 810 keV;

t_l_—“live” measurement time, s;

η—the yield of gamma quanta;

K—the correction for the geometry and self-absorption of gamma radiation in the source material, K = 1.03 for E = 510,810 keV;

ε—the registration efficiency;

λ—the decay constant of the nuclide formed in the AI, λ (^64^Cu) = 1.52 × 10^−5^ c^−1^, λ (^58^Co) = 1.13 × 10^−7^ c^−1^;

t_e_—the AI exposure time, s.

The efficiency of gamma-quanta registration was determined using calibration sources Eu-152, Cs-137, Na-22, Co-60 from the Exemplary gamma radiation sources (EGRS) set.

Measurements of the gamma spectra of the EGRS for determining the registration efficiency and for determining the activity of the AI were carried out in the same geometry. The registration efficiency was calculated using Equation (2):(2)$$\varepsilon (E)=\frac{S\left(E\right)}{t_{l}A_{0}\eta e^{-\lambda t_{e}}},$$
where, S(E) is the area of the peak of total absorption with energy E;

t_l_—“live” measurement time, s;

A_0_—activity of the EGRS at the time of certification, Bq;

η—the yield of gamma quanta;

λ—the decay constant of the radionuclide;

t_e_—the time elapsed since the certification of the OSGI, s.

Table 3 presents the radiation characteristics of the radioisotopes used.

The reaction rate R, related to one nucleus of the target isotope, was determined by the activity of the irradiated AI A_t_ and was reduced to the moment of completion of the launch (Equation (3)):(3)$$R=\frac{A_{t}}{N\cdot \left(1-e^{-\lambda t_{0}}\right)}$$
where, N = m·ν·N_A_/M—number of nuclei of the target isotope in the AI; m—mass of AI; ν—target isotope content in natural mixture, M—molar mass of target isotope;

λ—decay constant of a nuclide formed in an AI,

λ(^64^Cu) = 1.52·× 10^−5^ c^−1^,

λ(^58^Co) = 1.13·× 10^−7^ c^−1^;

t_0_—effective irradiation time at the start, t_0_ = 1860 s.

The reaction rate is the initial value for subsequent calculations of the neutron field characteristics.

As a result of the studies, the reaction rates of ^63^Cu(n,γ)^64^Cu were determined at the locations of the AI installation in the upper part of the IVG.1M reactor FEC after the above-mentioned IVG1M reactor start-up modes.

Table 4 shows the main results of the studies with copper wire AIs under all reactor irradiation modes of polymer samples with fillers.

Based on the analysis of the data presented in Table 4 of the experimental results, the coefficient of neutron absorption capacity of the composite material was determined for different reactor operating modes by dividing the value of the reaction rate of the internal AE by the value of the reaction rate of the external AE. The values of the neutron absorption coefficient are given in Table 5.

As can be seen from Table 5, the neutron absorption coefficient for this sample, regardless of the radiation dose and the integral radiation power, is within 7.5, which indicates the stability of the neutron absorption coefficient.

### 3.4. Results of X-Ray Phase Analysis

In order to assess the effect of reactor irradiation on the structure, phase composition and properties of radiation-protective composite material based on UHMWPE with fillers Pb, B, W (type B20W20Pb10), a set of materials science tests was carried out on samples, the markings of which are indicated in Table 6.

The following compounds are present in the sample: UHMWPE—50%, WC—20%, PbO—10%, H_3_BO_3_—20%.

UHMWPE is an ultrahigh molecular weight polyethylene (molecules: –CH_2_–CH_2_–).

WC—tungsten carbide, α-WC [20]—hexagonal syngony. spatial group P6-m2.

PbO—It is a lead oxide compound.

H_3_BO_3_—boric acid, in this modification, corresponds to the highest degree of boron oxidation, layered triclinic lattice.

Figure 8 shows a comparison of the peaks of diffractograms of UHMWPE samples with fillers before and after reactor irradiation.

The result of the phase identification of the peaks of the diffraction patterns of type 2 samples is shown in Figure 9. The inset shows an enlarged region of the diffraction patterns in the range of angles from 14 to 29°2θ.

The basis of the phase composition of type 2 samples is the lead oxide phase PbO_2_ of the tetragonal syngony.

The second phase in terms of peak intensity is the UHMWPE phase, identified by the n-hexatetracontane (C_46_H_94_) card of the orthorhombic syngony. The main consequences of 2θ for UHMWPE: the most intense 21.4220 with indices (110), the second along strike 23.7853 with indices (200).

The third in terms of peak intensity is the phase of higher tungsten carbide WC.

Peaks with low intensity are identified by the hydrogen borate card.

After irradiation in modes 1 and 2, an increase in the intensity of the UHMWPE peaks by about 2 times was noted, after irradiation in mode 3—by 1.5 times. The intensity level of the lead oxide peaks did not change after irradiation.

### 3.5. Results of SEM Analysis

After conducting a set of studies, the structure of irradiated and non-irradiated UHMWPE samples with filler was studied. Figure 10 shows the structure of a fragment of the composite material before irradiation in the IVG 1M reactor. In addition to the absence of cracks and other surface defects, no visible pores were found in the samples. This indicates the high quality of the obtained composite materials and confirms their integrity after the experiments. As can be seen from the figure, for the composite material with the addition of fillers, no signs of particle agglomeration are observed. The filler is presented in the form of individual scaly plates that are uniformly distributed in the UHMWPE matrix, which confirms the high homogeneity of the material structure. Such distribution indicates high-quality retention of filler particles in the composite matrix. At the same time, filler particles do not act as crack initiation centers, which indicates the stability and high strength of the material.

Figure 11 shows the structure of the composite material after irradiation in the IVG1M reactor at an integral power of 3 MW, shown at different magnifications. As can be seen, the structure of the material remains homogeneous, and the filler particles are clearly distinguishable. Reactor irradiation did not lead to visible changes in the structure of the composite. At a higher magnification, shown in Figure 11b, one can examine in detail the structure of the polymer itself. It is evident that the polymer matrix did not undergo any significant changes under the influence of irradiation in this mode, which confirms its stability under the selected parameters.

Figure 12 shows the structure of the composite material after irradiation at the IVG1M reactor with an integral power of 11.83 MW, shown at different magnifications. As can be seen, the structure of the composite material is also preserved without any noticeable changes after irradiation in this mode. H_3_BO_3_ particles are clearly visible, as well as the matrix of the binding polymer material. This indicates the stability of the composite and the absence of structural damage, despite the impact of high irradiation power.

Figure 13 shows the structure of the composite material after irradiation at the IVG 1M reactor, performed during the 3rd and 4th starts of the reactor, at various magnifications. It follows from the image analysis that the composite material has not undergone significant changes under the specified irradiation regime.

The absence of filler particle peeling indicates excellent adhesion between the filler particles and the polymer matrix, while thin nanofibril threads are observed on the surface of the samples. The formation of such structures confirms the adhesion between the filler and the matrix. The strength of these bonds exceeds the yield strength of the polymer, and when a load is applied to the composite, the interphase boundary is not destroyed, which would be observed in the case of poor adhesion, but the deformation of the polymer microvolumes begins and as a result, thin fibrous structures are formed—nanofibrils, at some point, the fibrils are torn away from the main volume of the matrix material, but the bonds with the filler are not broken, as a result of which a structure of nanofibrils remains on the surface of the filler particles.

## 4. Conclusions

As a result of the calculations carried out, the stationary temperature field of the model of an experimental device with irradiation samples on the outer surface of the PEC of the IVG.1M reactor was determined. According to the calculation results, the heating of the samples during the experiment was no more than 40 °C.

Physical studies with activation indicators were conducted on a series of IVG.1M reactor starts. Wire AIs were installed inside and outside the composite material samples. As a result of the studies, the density of the thermal and fast neutron flux for a composite material sample was determined after a series of starts with different integral powers. It was also established that for this composite material the neutron absorption coefficient is 7.5.

The results of X-ray phase analysis of the sample showed the presence of boric acid phase, beta lead oxide phase, phase responsible for the polyethylene structure, and a small content of tungsten carbide phase. After irradiation in modes 1 and 2, an increase in the intensity of UHMWPE peaks by approximately 2 times was noted after irradiation in mode 3—by 1.5 times. The intensity level of lead oxide peaks did not change after irradiation.

According to the results of SEM analysis of the composite material structure after various irradiation modes at the IVG1M reactor, it was found that after irradiation, the composite material did not undergo any changes. There is no chipping of filler particles, which indicates good adhesion between the filler particles and the polymer matrix, while thin threads of nanofibrils are visible on the surface. The formation of such structures confirms the adhesion between the filler and the matrix. The strength of these bonds exceeds the yield strength of the polymer, and when a load is applied to the composite, the interphase boundary does not break, which would be observed in the case of poor adhesion, but the deformation of the polymer microvolumes begins, and as a result, thin fibrous structures are formed—nanofibrils. At some point, the fibrils are torn from the main volume of the matrix material, but the bonds with the filler are not broken, as a result of which a structure of nanofibrils remains on the surface of the filler particles.

This section is not mandatory but can be added to the manuscript if the discussion is unusually long or complex.

## 5. Patents

Based on the results of the work carried out, we received a utility model, patent no. 7206, dated 3 March 2022: Powdered material for thermal spraying of polymer coatings. Ocheredko I.A., Skakov M.K., Tuyakbaev B.T., Erbolatuly D., Bayandinova M.B.

## Figures and Tables

**Figure 1 polymers-16-03425-f001:**
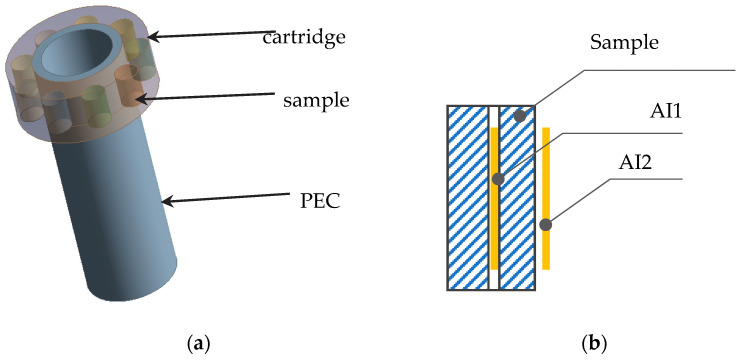
Scheme of sample placement in a cartridge case (**a**) and scheme of the AI placement in the studied samples (**b**).

**Figure 2 polymers-16-03425-f002:**
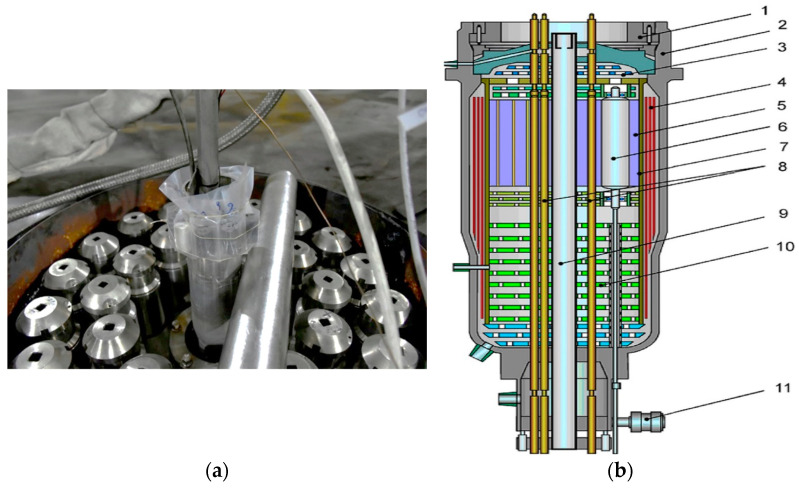
Cartridge belt with test samples on the reactor lid (**a**) and vertical section of the IVG.1M reactor (**b**). 1—Reactor lid; 2—Housing; 3—End screen; 4—Side screen; 5—Reflector; 6—Control drums (CD); 7—Central assembly; 8—Water-cooled process channels (WCPC) with low-enriched uranium fuel; 9—Experimental physical channel (EPC); 10—Iron-water shield; 11—CD drives.

**Figure 3 polymers-16-03425-f003:**
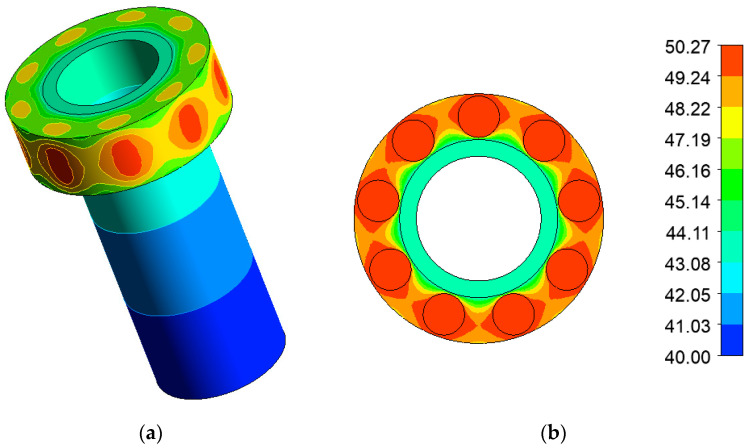
Calculated temperature field of the model (**a**), and the calculated field of the cross-section of the cartridge belt at the level of the middle of the height of the sample (**b**).

**Figure 4 polymers-16-03425-f004:**
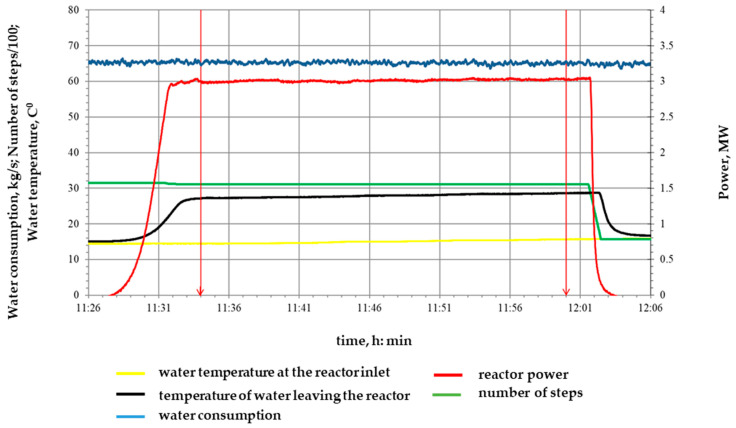
Diagram of the first realized launch of the IVG.1M research reactor for irradiation of samples.

**Figure 5 polymers-16-03425-f005:**
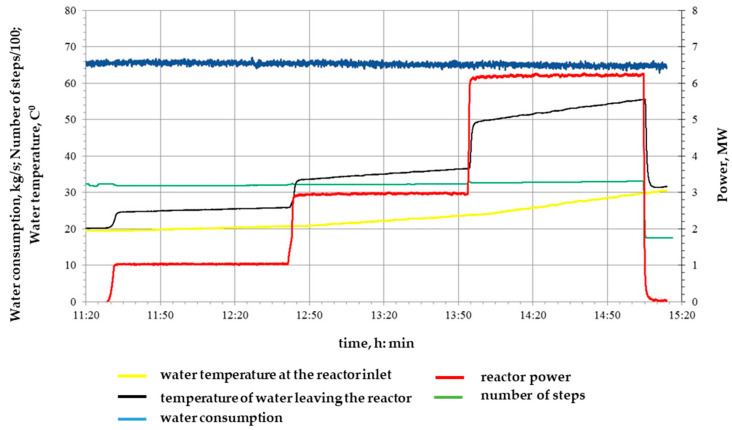
Diagram of the second completed launch of the IVG.1M research reactor for irradiation of samples.

**Figure 6 polymers-16-03425-f006:**
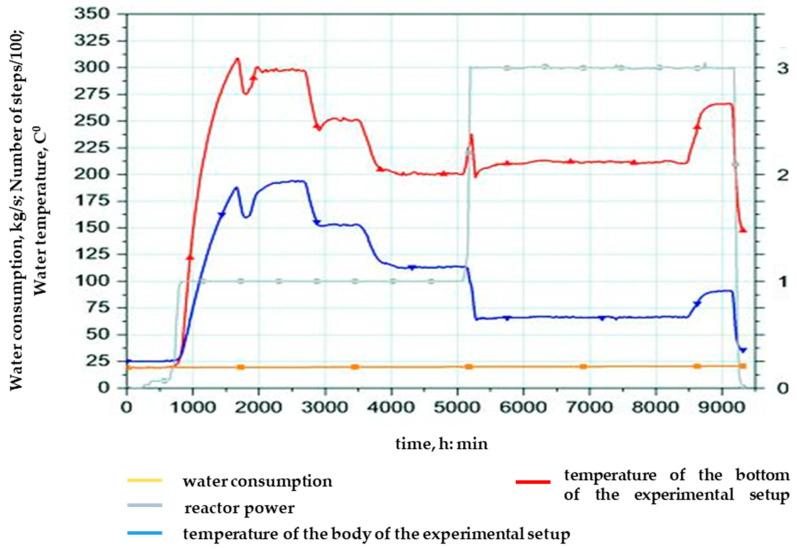
Diagram of the third realized launch of the IVG.1M research reactor for irradiation of samples.

**Figure 7 polymers-16-03425-f007:**
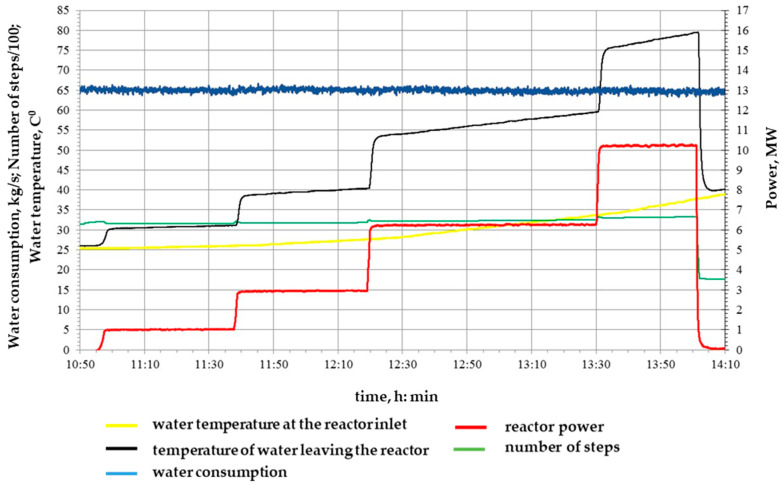
Diagram of the fourth realized launch of the IVG.1M research reactor for irradiation of samples.

**Figure 8 polymers-16-03425-f008:**
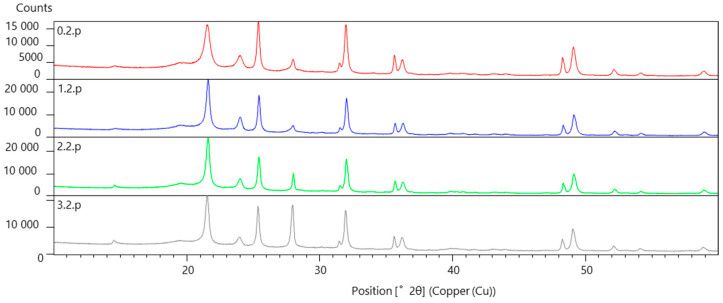
Diffraction patterns of UHMWPE in the initial state (red profile) and after irradiation (blue, green and gray profiles), Kα1.2 radiation.

**Figure 9 polymers-16-03425-f009:**
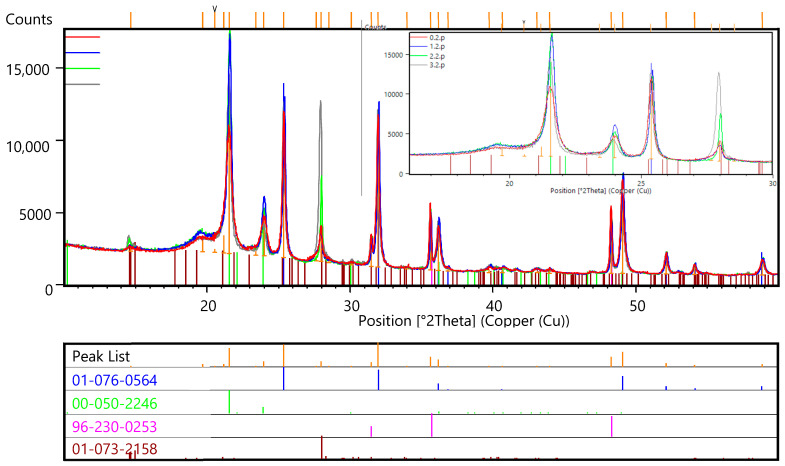
Result of processing the diffraction patterns of UHMWPE type 2 before (red profile) and after irradiation (blue, green and gray profiles), Kα1 radiation.

**Figure 10 polymers-16-03425-f010:**
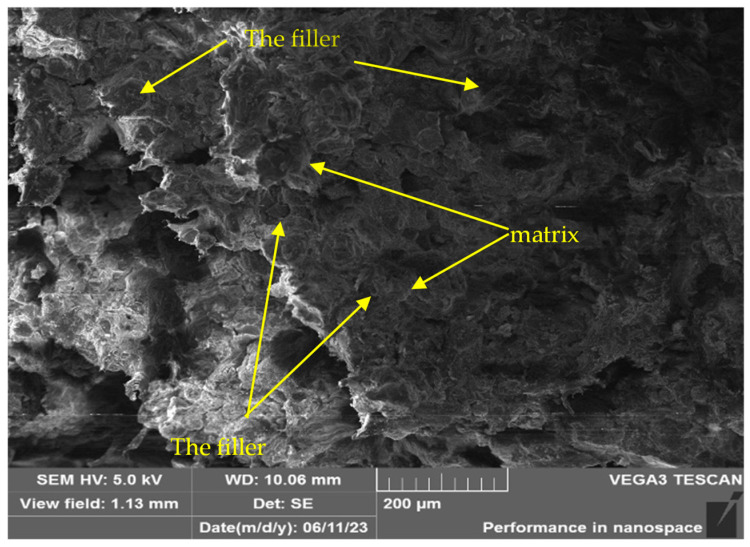
Structure of the composite material type B20W20Pb10 before irradiation at the IVG1M reactor.

**Figure 11 polymers-16-03425-f011:**
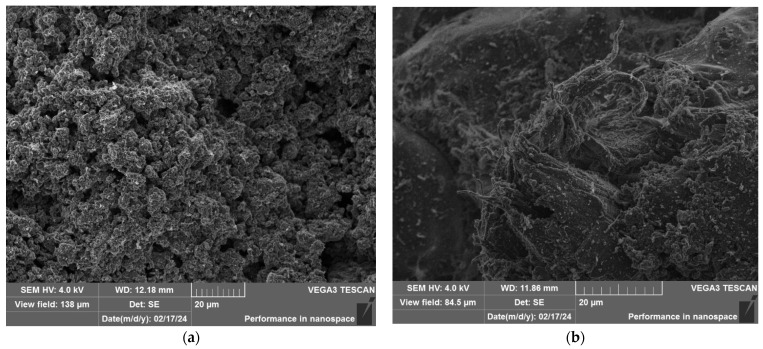
Structure of the composite material after irradiation at the IVG1M reactor with (**a**) an integral power of 3 MW at different magnifications (**b**).

**Figure 12 polymers-16-03425-f012:**
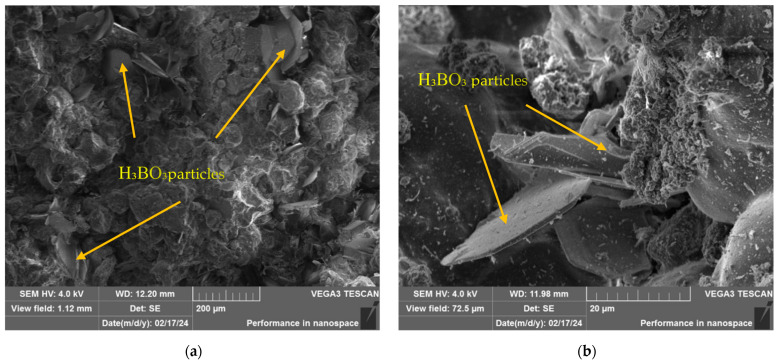
Structure of the composite material after irradiation at the IVG1M reactor with (**a**) an integral power of 11.83 MW at different magnifications (**b**).

**Figure 13 polymers-16-03425-f013:**
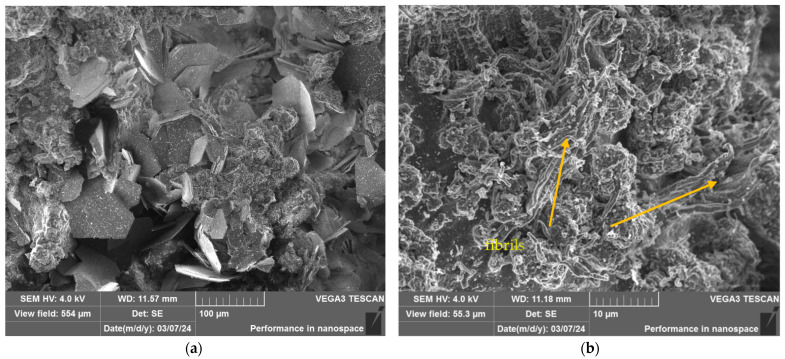
Structure of the composite material after irradiation at the IVG1M reactor after the 3rd (**a**) and 4th start-up of the IVG1M reactor at different magnifications (**b**).

**Table 1 polymers-16-03425-t001:** Chemical composition of samples.

Type No	Base Material	Conventional Name	Content of Additives, % by Weight			Dimensions, mm
			H_3_BO_3_	WC	PbO	
1	Ultra-high molecular weight polyethylene	B20W20Pb10	20	20	10	Ø20 × 25

**Table 2 polymers-16-03425-t002:** Main technical characteristics of the IVG.1M reactor.

Parameter	Value
Fuel enrichment, %	19.75
Thermal power, MW	
– design	72
– operational	10
Neutron flux density, n/cm^2^s	
– fast	4.0 × 10^12^
– thermal	2.5 × 10^13^
Reactor core size, mm	
– effective diameter	548
– height	800
Number of EPC, pcs	1

**Table 3 polymers-16-03425-t003:** Characteristics of radioisotopes.

	Parameter	Half-Life	E_1_, keV (Output, %)	E_2_, keV (Output, %)	E_3_, keV (Output, %)
Isotope					
^58^Co		70.92 day	810 (0.99)	-	-
^64^Cu		12.7 h	511 (29)	1345 (0.45)	-
EGRS					
^22^Na		2602 years	511 (181)	1274.5 (99.94)	-
^60^Co		5.27 years	1173.2 (99.9)	1332.48 (99.98)	-
^137^Cs		30.0 years	661.6 (85)	-	-
^152^Eu		13.33 years	244 (7.52)	344 (26.58)	1408 (20.8)

**Table 4 polymers-16-03425-t004:** Characteristics of radioisotopes.

Start Number/ Place of Installed AI	Mass of AI, mg	Activity ^64^Cu, Bq/g(AI)	Reaction Speed·10^−15^, Reaction/(s·Nucleus)
1 start internal AI	126.0	7.9 × 10^4^	0.44
1 external start AI	140.0	3.8 × 10^5^	2.07
2 start internal AI	85.4	6.8 × 10^4^	0.52
2 external start AI	156.3	5.1 × 10^5^	3.9
3 start internal AI	105.3	0.27	0.51
3 external start AI	138.6	1.95	3.70
4 start internal AI	105.3	0.89	0.54
4 external start AI	138.6	6.9	4.2

**Table 5 polymers-16-03425-t005:** Neutron absorption coefficients of the B20W20Pb10 sample after all reactor irradiation modes.

Start Number	Integral Irradiation Power, MW	Integral Radiation Dose	Neutron Absorption Coefficient of Samples
1	3	7.2 × 10^15^ n/cm^2^	7.75
2	11.83	5.1 × 10^16^ n/cm^2^	7.5
3	4.73	3.4 × 10^16^ n/cm^2^	7.25
4	15	4.4 × 10^16^ n/cm^2^	7.78

**Table 6 polymers-16-03425-t006:** Sample marking.

Type UHMWPE	Irradiation Mode	Designation of the Diffractogram
B20W20Pb10	Without irradiation	0.2.p
	1 start	1.2.p
	2 start	2.2.p
	3 + 4 starts	3.2.p

## Data Availability

Data will be made available on request.

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
