# Peer review of "The Neutron Absorption Capacity of a Composite Material Based on Ultrahigh Molecular Weight Polyethylene Under Reactor Radiation Conditions"

_polymers, 2024, doi:10.3390/polym16233425_

Round 1

Reviewer 1 Report

Comments and Suggestions for Authors

Dear authors:

The manuscript presents the results of an interesting study on the influence of fillers on the neutron absorption capacity of materials made from ultra-high molecular weight polyethylene. It is based on scientific principles and is well-developed.

In any case, I believe that before its publication, the manuscript should be carefully reviewed to correct some potential errors. For example:

- in lines 16-17, a nomenclature is used, 63Cu (n, g), 64Cu 16 and 58Ni (n, p) 58Co, which in other parts of the article is indicated with superscripts and spacing;

- throughout the text, there are units where subscripts and superscripts have not been used correctly (for instance, see line 86 for the term mol-1, and in Table 2, n/cm2s);

- in lines 101 and 102, percentages are cited that exceed 100% by a significant margin (if corrected, it should also be reflected in Table 1);

- in many of the graphs, at least in my PDF version, the labels and units on the axes are not clearly visible.

In summary, I believe this is a study that deserves to be published in Polymers, but only after a careful review of the style and certain details.

All the best,

Author Response

comment 1 - lines 16-17 use the nomenclature 63Cu(n,g), 64Cu16 and 58Ni(n,p)58Co, which in other parts of the article is indicated by superscripts and spaces – corrected

comment 2 - in the text there are units in which subscripts and superscripts are used incorrectly (for example, see line 86 for the term mol-1 and in table 2 n/cm2s) – corrected

comment 3- lines 101 and 102 indicate percentages significantly exceeding 100% (in case of correction, this should also be reflected in table 1) – Thanks for the note, we edited the composition in accordance with the experimental data, an error was made at the stage of translating the article into English language

- on many graphs, at least in my PDF version, the labels and units of measurement on the axes are not clearly visible - corrected in accordance with the comment, offsets were made when converting the Word document

Reviewer 2 Report

Comments and Suggestions for Authors I would like to point out that I am not an expert in materials capable of
absorbing/slowing down neutrons, but my skills are in the chemistry of materials. For this reason my comments refer only to the second part of the work.
the materials are certainly very interesting but I think their characterization
needs to be implemented. In particular, the results of the XRD study should be better explained by also
indicating the values ​​of 2 theta and seeking an explanation for the changes
in the polymer signals.
In my opinion some FTIR spectra to study the polymer before and after the
irradiation must be detected. Furthermore, some experiments to evaluate the
mechanical properties could be carried out.
In differen parts of the paper the amounts of boric acid and WC are different.
Which is the true composition? 70% polymer + H3BO3 10 or 20% + PbO 10 or 20% +
WC 20%. How was it calculated being the sum is greater than 100% ?

Author Response

comment 1 - In particular, the results of the XRD study should be better explained by also specifying the 2 theta values ​​and attempting to explain the changes in the polymer signals.

Thanks for your comments, we have edited some data and added 2 theta values ​​with indices for UHMWPE. To fully explain the changes in UHMWPE peak intensity before and after reactor irradiation, we need to conduct additional studies. Some work involving research with irradiated materials requires time to reduce radiation activity in a hot chamber where it is not possible to carry out all types of analysis, however, we will certainly take into account your comments when performing more thorough studies on X-ray phase analysis of UHMWPE before and after reactor irradiation in our next works.

comment 2 - In my opinion, it is necessary to detect some FTIR spectra to study the polymer before and after irradiation.

Since at the moment the project for which the work is being funded lasts 3 years, some work involving research with irradiated materials requires time to reduce radiation activity in a hot chamber where it is not possible to carry out all types of analysis, and therefore at the moment we only have data from the original samples, but we will certainly provide the work you indicated in our next works

comment 3- In addition, some experiments could be carried out to evaluate the mechanical properties.

Since at the moment the project for which the work is being funded lasts 3 years, some work involving research with irradiated materials requires time to reduce radiation activity in a hot chamber where it is not possible to carry out all types of analysis, and therefore at the moment we only have data from the original samples, but we will certainly provide the work you indicated in our next works.

comment 4 - The amounts of boric acid and WC vary in different parts of the article. What is the true composition? 70% polymer + H3BO3 10 or 20% + PbO 10 or 20% + WC 20%. How was this calculated if the amount is greater than 100%?

Corrected throughout the entire manuscript, our young specialists made a mistake when forming the manuscript, since we prepared a total of more than 50 samples for this work.

Reviewer 3 Report

Comments and Suggestions for Authors

1. In the end of Introduction, please describe more clear the main aims of the paper and the tasks that will be fulfilled.

2. If possible, add results from Raman spectroscopy.

3. Please, correct the references according to journal requirements. Add DOI.

Author Response

comment 1 - At the end of the introduction, please describe more clearly the main goals of the article and the tasks that will be accomplished.

Added

comment 2 - If possible, add Raman spectroscopy results.

Since at the moment the project for which the work is being funded lasts 3 years, some work involving research with irradiated materials requires time to reduce radiation activity in a hot chamber where it is not possible to carry out all types of analysis, and therefore at the moment we only have data from the original samples, but we will certainly provide the work you indicated in our next works

comment 3 - Please correct the references as required by the journal. Add DOI.  Thanks for your comments.

Added DOIs.